# Predictive Factors of Ventilatory Support in Chest Trauma

**DOI:** 10.3390/life11111154

**Published:** 2021-10-29

**Authors:** Silvia Fattori, Elisa Reitano, Osvaldo Chiara, Stefania Cimbanassi

**Affiliations:** 1General Surgery-Trauma Team, Niguarda Hospital, 20162 Milan, Italy; 2Division of General Surgery, Department of Translational Medicine, Maggiore della Carità Hospital, University of Eastern Piedmont, 28100 Novara, Italy; elisa.reitano@live.it; 3Department of Pathophysiology and Transplants-State, University of Milan-General Surgery-Trauma Team, Niguarda Hospital, 20162 Milan, Italy; osvaldo.chiara@ospedaleniguarda.it

**Keywords:** trauma, chest trauma, thorax injury, non-invasive ventilation, invasive ventilation, outcome

## Abstract

This study aims to define possible predictors of the need of invasive and non-invasive ventilatory support, in addition to predictors of mortality in patients with severe thoracic trauma. Data from 832 patients admitted to our trauma center were collected from 2010 to 2017 and retrospectively analyzed. Demographic data, type of respiratory assistance, chest injuries, trauma scores and outcome were considered. Univariate analysis was performed, and binary logistic regression was applied to significant data. The injury severity score (ISS) and the revised trauma score (RTS) were both found to be predictive factors for invasive ventilation. Multivariate analysis of the anatomical injuries revealed that the association of high-severity thoracic injuries with trauma in other districts is an indicator of the need for orotracheal intubation. From the analysis of physiological parameters, values of systolic blood pressure, lactate, and Glasgow coma scale (GCS) score indicate the need for invasive ventilatory support. Predictive factors for non-invasive ventilation include: RTS, ISS, number of rib fractures and presence of hemothorax. Risk factors for death were: age over 65, the presence of bilateral rib fractures, pulmonary contusion, hemothorax and associated head trauma. In conclusion, the need for invasive ventilatory support in thoracic trauma is associated to the patient’s systemic severity. Non-invasive ventilation is a supportive treatment indicated in physiologically stable patients regardless of the severity of thoracic injury.

## 1. Introduction

Trauma represents one of the most important causes of mortality and morbidity worldwide [1,2]. Chest trauma represents the third cause of death in 20–40% patients with multisystemic trauma, following head and spinal injuries [3,4]. Rib fractures, pneumothorax, flail chest and pulmonary contusion are the most common injuries caused by thoracic trauma [5]. After a chest trauma, the lungs are compromised by direct parenchymal damage and systemic activation of inflammatory system that causes dysfunction of the alveolar-capillary barrier. As a consequence of these alterations hemorrhage, extravasation of fluid from the vascular bed and pulmonary destruction occur. These events associated with increased alveolar secretions and airways colonization promote alveolar collapse, ventilation-perfusion shunt, consolidation, and pneumonia, all leading to hypoxemia and acute respiratory disease [6,7]. These complications are associated with the development of acute lung failure in up to 20% patients with thoracic trauma [4].

Therefore, ventilation plays a main role in the treatment of patients with thoracic trauma to prevent the onset of the abovementioned complications. Nowadays, there are no guidelines regarding the use of invasive and non-invasive ventilation in trauma patients although the application of these interventions is increasing in the clinical practice [8]. The aim of this study is to define the predictive indicators of the need for ventilatory support in patients with thoracic trauma.

## 2. Materials and Methods

Data about all patients with thoracic trauma admitted to the Niguarda Trauma Center from October 2010 until November 2017 were extracted from our Trauma Registry and retrospectively analyzed. The Trauma Registry is constantly and prospectively updated by a Trauma Team consultant, and annually revised by the Head of Department.

Demographic data, abbreviated injury scale (AIS, 1998 version) of each anatomic region, injury severity score (ISS), revised trauma score (RTS), the probability of survival (PS) obtained from the TRISS system (trauma and injury severity score), length of hospitalization (LOS) and outcome were evaluated. Subsequently, the American Society of Anesthesiologists (ASA) physical status classification was chosen to summarize comorbidities in an ordinal way. Rib fractures were divided into five groups in order to stratify the chest trauma severity: less than two fractured ribs, between three and five, higher or equal to six, fractures involving the first and/or second ribs and bilateral rib fractures. Any further injuries to the chest were included: heart or major vessel injuries, pneumothorax, hemothorax, hemopneumothorax, pulmonary contusion, fracture of the scapula, clavicle or sternum and pleural effusion.

Patients were divided into groups based on the type of ventilation to which they were subjected: spontaneous breathing, invasive ventilation, non-invasive ventilation. Patients were also divided into two groups based on the outcome: deceased or survived.

Retrieved data were recorded in a computerized spreadsheet (Microsoft Excel 2016; Microsoft Corporation, Red Mond, WA, USA) and analyzed with statistical software (IBM Corp., released 2020, IBM SPSS Statistics for Windows, version 27.0; Armonk, NY, USA, IBM Corp.).

The distribution of the sample was evaluated with the Kolmogorov–Smirnov and Shapiro–Wilk tests resulting in a non-normal distribution for any of the variables examined. Categorical variables were compared using Pearson′s Chi-square test or Fisher′s test for small samples, numerical variables were studied by comparing medians and the Mann–Whitney test. A *p*-value ≤ 0.05 was considered statistically significant for all tests.

Multivariate analysis was used to provide odds ratio with a 95% confidence interval for the individual variables, identifying possible predictors of intubation, non-invasive ventilation, and mortality. Different multivariate regression models were built: one for invasive ventilation, one for non-invasive ventilation and one for mortality. Subsequently, for the invasive treatment group, two further models were constructed, one exclusively for anatomical injuries and one for physiological alterations.

The Trauma Registry of the Trauma Center of the Niguarda Hospital in Milan was formerly approved for scientific purposed by the Niguarda Milano Area Three Ethics Committee (Record Number 534-102018). Given the retrospective nature of the study, no specific approval from the ethics review committee was required.

## 3. Results

The study included 832 patients who arrived at the Trauma Center of the Niguarda Hospital in Milan; 653 were male (78.50%) and 179 were female (21.51%). The age of the patients had a median of 49 years (IQR1 = 38, IQR3 = 62). The age groups were divided according to the previous and most recent WHO classification with a threshold of 65 and 75 years respectively: 181patients were over 65 (21.7%) and 94 over 75 (11.3%).

The ASA scale was considered in this study for the evaluation of comorbidities; 508 patients (60.90%) had an ASA score of one, 266 (31.90%) had an ASA score of two, 55 (6.60%) had an ASA score of three, finally 2 (0.20%) patients had an ASA score of four.

The dynamics of the trauma were classified into nine different categories: motorcycle accident involving 256 patients (30.70%), precipitation involving 189 patients (22.70%), car accident involving 161 patients (19.30%), pedestrian hit involving 121 patients (14.50%), bicycle accident involving 57 patients (6.80%), crushing involving six patients (0.70%), stab wounds involving only one patient (0.10%), gunshot wounds involving only one patient (0.10%). For 42 patients, it was not possible to trace the mechanism of the trauma.

The pre-hospital data were analyzed: RTS had a median of 12 (IQR1 = 11, IQR3 = 12), oxygen saturation a median of 98% (IQR1 = 95, IQR3 = 99), the GCS a median of 15 (IQR = 13, IQR = 15). Among the data collected in the emergency room: RTS had a median of 12 (IQR1 = 4, IQR = 12), oxygen saturation a median of 99% (IQR1 = 96, IQR3 = 100), the GCS (IQR1 = 3, IQR3 = 15) a median of 15 and systolic blood pressure a median of 130 mmHg (IQR1 = 110, IQR3 = 145).

Regarding the blood tests performed, lactate values presented a median of 2.4 (IQR1 = 1.64, IQR3 = 3.5), base excess (BE) values a median of −2.4 (IQR1 = −5.25, IQR3 = 0.4) and the test for ethanolemia was positive for 102 patients (12.20%).

Considering imaging used in the emergency room, 188 patients (22.50%) had tested positive for chest injuries on E-FAST 691 patients (82.90%) were evaluated by chest X-ray and 732 patients (87.80%) by torso CT.

Twenty-eight patients (3.40%) underwent decompressive mini-thoracotomy, 115 patients (13.80%) underwent right chest tube thoracostomy placement and 118 (14.10%) left chest tube thoracostomy. Finally, 312 patients (37.40%) required emergency surgery.

Among the injuries found; the median number of rib fractures was four (IQR1 = 2, IQR3 = 7). Patients with less than two fractured ribs were 256 (30.70%), 296 patients (35.50%) had three to five fractured ribs, 280 (33.60%) had more than six fractured ribs. Finally, the patients presenting with fractures at the level of the first and/or the second rib were 322 (38.60%) and 180 (21.60%) presented with bilateral rib fractures. Other injuries found were: injuries to the heart or large vessels of the thorax in 19 patients (2.3%), pneumothorax in 326 patients (39.10%), hemothorax in 42 patients (5%), hemopneumothorax in 84 patients (10.10%), pulmonary contusion in 394 patients (47.20%), scapula, clavicle, or sternum fractures in 134 patients (16.10%) and pleural effusion in 142 patients (17%).

A total of 166 patients (19.90%) had isolated thoracic trauma while 666 (79.90%) had thoracic trauma associated with one or more additional traumas. Three hundred-and-eighty-six patients (46.30%) had an abdominal trauma, 526 (63.10%) head injury and 572 (68.60%) trauma to the extremities.

The different trauma scores were analyzed, in particular the AIS of the chest had a median of three (IQR1 = 3, IQR3 = 4), the AIS of the abdomen a median of two (IQR1 = 2, IQR3 = 3), AIS of the head a median of two (IQR1 = 2, IQR3 = 4), the AIS of the extremities a median of two (IQR1 = 2, IQR3 = 4), the ISS a median of 24 (IQR1 = 14, IQR3 = 36) and the death probability had a median of 4.8 (IQR1 = 1.6, IQR3 = 19).

Finally, 77 (9.20%) patients deceased, 336 (40.30%) underwent orotracheal intubation and 181 (21.70%) treated with non-invasive ventilation. These population data are shown in Appendix A in Appendix A. 

The group of patients who underwent invasive ventilation was compared to the control group by statistical univariate analysis (Table 1 and Table 2). From this comparison, numerous statistically significant variables emerged. Oxygen saturation and RTS both detected both in pre-hospital and in the emergency room, BE, lactate values, necessity of decompressive mini-thoracotomy, number of rib fractures, presence of first and/or second rib fractures, bilateral rib fractures, heart or large vessels injuries, pneumothorax, hemothorax, hemopneumothorax, pulmonary contusion, scapula, clavicle or sternum fractures, pleural effusion, ISS and associated traumas in different areas than the chest resulted with a *p*-value ≤ 0.05 and were analyzed with multivariate regression.

The significant variables were then analyzed with binary logistic regression. Based on the characteristics of the test—that does not allow to include more than a certain number of variables—and to avoid data repetitions it was decided to not incorporate some variables in the analysis. Systolic blood pressure and GCS were excluded as they are already represented within the RTS score since it is calculated by the sum of these parameters. The AIS of chest, abdomen, head, and extremities were not included as they are already represented in the ISS score and defined by the presence of trauma in these locations. Furthermore, it was decided to include only the number of fractured ribs and not differentiate them into groups (rib fractures ≤2, 3–5, ≥6). Finally, the placement of right and left chest tube thoracostomy and the need for surgery were not considered in the analysis since chest tube is often positioned as a function of the need of invasive ventilation support and surgery always requires intubation.

The binary logistic regression (Table 3) showed only two predictive factors of the need for invasive ventilatory support: the value of RTS and ISS. The scores had respective *p*-values of <0.001 and 0.003 and odd’s ratio (OR) of 1.092 and 0.329 (CI = 95%). It is important to note that the OR of the RTS is correctly lower than one, as the value of this score decreases as the severity of the altered of the parameters increases. Since the two scores describe anatomical and physiological alterations respectively; they were evaluated separately.

The analysis of anatomical injuries revealed as statistically significant data the presence of bilateral rib fractures, hemothorax, hemopneumothorax, pulmonary contusion and the association of thoracic trauma with abdominal, head and extremity trauma. The values are shown in Table 4.

In the multivariate model for the physiological alteration (Table 5), the values of GCS and systolic blood pressure in the emergency room and the lactates values resulted statistically significant, with an OR of 0.366, 0.973 and 1.221 (CI = 95%), respectively. The OR of GCS and blood pressure is lower than one as the values of these parameters decrease as the alterations increase.

Data from the group of patients who underwent non-invasive ventilation was compared to the control group. From this survey, a large number of significant variables were found, shown in Table 6 and Table 7.

Among the findings of several variables with *p*-value > 0.05 were found: gender, age, ASA scale, oxygen saturation detected in emergency room, blood alcohol test positivity, the presence of heart or large vessels injuries, hemopneumothorax and head trauma.

The significant variables just described were studied with binary logistic regression (Table 8). Some variables were not included for the same reasons as for the multivariate analysis of invasive ventilation group: AIS of chest and head, the stratification of rib fractures (rib fractures ≤2, 3–5, ≥6), GCS and systolic blood pressure.

The outcome of this test revealed as predictors of need of NIMV: the number of fractured ribs, the presence of hemothorax, the values of ISS and RTS. The variables have significant *p*-values equal to 0.001, 0.040, 0.018 and 0.034 (CI = 95%) respectively; the OR were respectively 1.140, 2.962, 1.029 and 1.224 (CI = 95%).

Finally, it was decided to analyze the possible predictors of mortality in thoracic trauma. In the univariate analysis, numerous variables were found to be significant, represented in Table 9 and Table 10.

From the multivariate analysis (Table 11) the predictive factors for mortality after a chest trauma were: age over 65 years (*p*-Value < 0.001; OR = 5.146; CI = 95%), the presence of bilateral rib fractures (*p*-Value = 0.001; OR = 2.952; CI = 95%), hemothorax (*p*-Value = 0.018; OR = 3.133; CI = 95%) and pulmonary contusion (*p*-Value = 0.030; OR = 1.858; CI = 95%). Furthermore, the presence of head trauma was particularly significant (*p*-Value < 0.001; OR = 23.638; CI = 95%).

## 4. Discussion

From this retrospective analysis of patients admitted at a high-volume trauma center who suffered from thoracic trauma, it was possible to obtain various mortality and morbidity data. The following results emerged from the study:○Thoracic trauma is associated with the need for invasive ventilatory support in 40.30% and for non-invasive ventilatory support in 21.70%.○Patient severity is the main factor indicating the need for orotracheal intubation in thoracic trauma. Specifically, severity represented by the values of the RTS and ISS.○Regarding physiological parameters, modifications in the GCS score, systolic blood pressure and lactate values were predictive factors for the need for invasive ventilatory support.○Regarding anatomical injuries, pulmonary contusion, hemothorax, hemopneumothorax, bilateral rib fractures, the association with thoracic trauma of abdominal, head or extremity trauma were significant predictors for the need for invasive ventilation.○The presence of hemothorax, the increasing number of rib fractures and the ISS were instead predictive factors for the need for non-invasive ventilatory support.○Mortality in these patients was 9.20%. Age over 65, the presence of pulmonary contusion, hemothorax, bilateral rib fractures and head trauma have been shown to be risk factors for death.

The study is consistent with the literature indicating that trauma is a condition that predominantly affects men [9]. From the analysis of the dynamics of the trauma—although not available in all patients—it is possible to attribute to motorcycle accidents the main cause of traumatic events and of thoracic trauma, as reported in the literature [9,10].

The most common injuries detected were rib fractures, pneumothorax, and pulmonary contusion. As described by other studies, head and extremity injuries were the most associated to thoracic trauma [3,4].

Mortality in patients was 9.20%. Especially the age over 65, the presence of pulmonary contusion, hemothorax, bilateral rib fractures and head trauma have been shown to be risk factors for death.

The mortality detected in the group of patients in this study was consistent with that one described in the literature which is 10% [10,11]. As described above, the main risk factors for death were age over 65 years and head injury. These results are not surprising as since advanced age causes stiffening of the rib cage, exposing the patient to more serious injuries. In addition, head trauma is the main cause of death in patients with multisystemic trauma [12].

Thoracic trauma was associated with the need of invasive ventilatory support in 40.30% and of non-invasive ventilatory support in 21.70% of patients.

The severity of the patient was the main factor for orotracheal intubation in thoracic trauma, as supported from other different studies [13]. Specifically, the severity is represented by the values of the RTS and ISS. The ISS and the RTS are two scores that respectively describe the physiological state and the severity of the anatomical injuries.

Regarding physiological parameters, the GCS score, the systolic blood pressure, and lactate values emerged as indicators of need of invasive ventilation support. These parameters are descriptive of the patient′s neurological status and hemodynamic stability, respectively. It is, therefore, possible to conclude that in the presence of an alteration of one of these two physiological functions it is advisable to proceed with orotracheal intubation.

Concerning the severity of injuries, bilateral rib fractures, hemopneumothorax, hemothorax, and pulmonary contusion are high-severity thoracic injuries defined by a AIS of three or four, corresponding to potentially life-threatening injuries. These injuries and the association of trauma in locations other than the thoracic one, are indicators of the need for invasive ventilatory support.

It can be, therefore, concluded that the need for invasive ventilatory support is determined by the patient′s general clinical severity, defined by alterations in physiological parameters and severity of injuries.

Non-invasive ventilation is an important ventilatory support treatment as it is part of the therapy for respiratory failure. However, there are inconsistent guidelines in literature for its use in thoracic trauma, although the safety and the increasing application of NIMV has been assessed in several studies [8,14]. The importance of this treatment is also determined by fewer side effects than compared to invasive ventilation which is associated to higher rates of nosocomial pneumonia and prolonged mechanical ventilation [4,6,15].

This study demonstrated that the presence of hemothorax, the number of fractured ribs and the ISS were predictive factors for the need of non-invasive ventilatory support. Hemothorax and rib fractures are injuries with high risk of complication such as atelectasis and disarrangement in ventilation and gas exchange causing respiratory impairment and respiratory failure [4,16]. For this reason, this study has shown that non-invasive ventilation should be indicated in the presence of these injuries to improve respiratory function and promote proper lung expansion [4,16].

In this case, RTS describes a state of stability from a physiological point of view, thus defining non-invasive ventilatory support as the indicated treatment for physiologically stable subjects.

From these results it can be deduced that unlike invasive treatment, the indications for non-invasive ventilation are determined only by the presence of anatomical injuries and their severity in patients with stable parameters.

In conclusion, this study was aimed at identifying possible indicators of the need for ventilatory support. Considering the wide statistical sample of this study together with the lack of guidelines of ventilatory support in literature we believe this study could be a useful tool for clinicians and which lays the foundations for future prospective investigations.

## 5. Limitations

This study had some limitations, considering its retrospective nature as well as the inclusion criteria. Since all the patients with thoracic trauma were considered, we do not report selection bias. However, due to the retrospective nature of the study we report an information bias due to the lack of some data. In the future, it would therefore be interesting to perform a prospective study to better characterize the data according to the treatment.

## 6. Conclusions

Thoracic trauma is associated with the need for invasive ventilatory support rather than with local alterations, due to systemic clinical severity. More specifically, invasive ventilation should be used in all those patients who have hemodynamic instability and an alteration of consciousness. Orotracheal intubation should also be indicated in all those patients who have severe lesions in the chest, associated with trauma in other sites, which can affect normal ventilation predisposing the patient to a risk of hypoxia and therefore major systemic complications.

Finally, non-invasive ventilation is a supportive treatment that should be considered in all those patients who have stable physiological parameters but chest injuries that can compromise normal respiratory dynamics.

## Figures and Tables

**Table 1 life-11-01154-t001:** Invasive mechanical ventilation univariate analysis: demographic and trauma-related data.

Invasive Mechanical Ventilation												
		No IMV					IMV					
		n	%	IQR1	Median	IQR3	n	%	IQR1	Median	IQR3	*p*-Value
Gender	Female	112	22.58%				67	19.94%				0.39
	Male	384	77.42%				269	80.06%				
Age				39	50	63			38	49	62	0.273
Age > 65		111	22.38%				70	20.83%				0.609
Age > 75		59	11.92%				35	10.42%				0.577
ASA score	1	301	60.81%				207	61.61%				0.987
	2	160	32.32%				106	31.55%				
	3	33	6.67%				22	6.55%				
	4	1	0.20%				1	0.30%				
RTS pre-hosp ***				12	12	12			8	11	12	<0.001
SpO2 pre-hosp ***				96	98	99			91	97	99	<0.001
GCS pre-hosp ***				15	15	15			7	12	15	<0.001
RTS in ED ***				12	12	12			4	4	11	<0.001
SpO2 in ED				97	98	100			96	99	100	0.491
GCS in ED ***				15	15	15			3	3	10	<0.001
Systolic blood pressure in ED ***				120	135	150			90	110	130	<0.001
Mechanisms of injury trauma ***	other	6	1.26%				5	1.54%				0.001
	stab wound	1	0.21%				0	0.00%				
	gunshot wound	0	0.00%				1	0.31%				
	car accident	110	23.01%				51	15.69%				
	bike accident	39	8.16%				18	5.54%				
	motorcycle accident	162	33.89%				94	28.92%				
	pedestrian accident	67	14.02%				54	16.62%				
	precipitation	88	18.41%				101	31.08%				
	crusching	5	1.05%				1	0.31%				
BE ***				−3.2	−1.1	0.9			−7.1	−4.05	−1.1	<0.001
Lactate values ***				1.495	2.09	2.79			2.18	3.2	5.38	<0.001
E-FAST ***		80	16.43%				108	33.44%				<0.001
Chest X- ray ***		446	89.92%				245	72.92%				<0.001
Torso CT ***		418	84.27%				314	93.45%				<0.001
Decompressive mini-thoracotomy ***		1	0.20%				27	8.04%				<0.001
Right chest tube thoracostomy ***		36	7.26%				79	23.51%				<0.001
Left chest tube thoracostomy ***		27	5.44%				91	27.08%				<0.001
Emergency surgery ***		98	19.76%				214	63.69%				<0.001

*** Significant value.

**Table 2 life-11-01154-t002:** Invasive mechanical ventilation univariate analysis: injuries and trauma score data.

Invasive Mechanical Ventilation												
		No IMV					IMV					
		n	%	IQR1	Median	IQR3	n	%	IQR1	Median	IQR3	*p*-Value
n° rib fractures ***				2	3	6			3	5	8	<0.001
Rib fractures ≤2 ***		180	36.29%				76	22.62%				<0.001
Rib fractures 3–5		183	36.90%				113	33.63%				0.339
Rib fractures ≥6 ***		133	27.09%				147	43.75%				<0.001
1st and/or 2nd rib fractures ***		157	31.72%				165	49.11%				<0.001
Bilateral rib fractures ***		71	14.31%				109	32.44%				<0.001
Heart/large vessels injuries ***		4	0.81%				15	4.46%				0.001
Pneumothorax ***		168	33.87%				158	47.02%				<0.001
Hemothorax ***		15	3.02%				27	8.04%				0.002
Hemopneumothorax ***		27	5.44%				57	16.96%				<0.001
Pulmonary contusion ***		179	36.09%				215	63.99%				<0.001
Scapula/clavicle/sternum fractures ****		63	12.70%				71	21.23%				0.001
Pleural effusion ***		65	13.13%				77	22.92%				<0.001
Isoleted chest trauma ***		142	28.63%				24	7.14%				<0.001
Abdomen trauma ***		177	35.69%				209	62.20%				<0.001
Head trauma ***		268	54.03%				260	77.38%				<0.001
Extremities trauma ***		306	61.69%				267	79.46%				<0.001
Chest AIS ***				3	3	3			3	3	4	<0.001
Abdomen AIS ***				2	2	3			2	2	4	0.008
Head AIS ***				2	2	3			3	4	5	<0.001
Extremities AIS ***				2	2	3			2	3	4	<0.001
ISS ***				12	17	24			29	36	43	<0.001
Death probability ***				0.9	2.2	6.1			6.2	22.2	63.45	<0.001

*** Significant value.

**Table 3 life-11-01154-t003:** Invasive mechanical ventilation multivariate analysis: general variables.

Invasive Mechanical Ventilation						
	B	S.E.	Wald	gl	Sign.	Odd Ratio
SpO2 pre-hos	−0.053	0.043	1.487	1	0.223	0.949
RTS pre-hosp	0.517	0.336	2.336	1	0.124	0.596
RTS in ED ***	−1.112	0.371	8.974	1	0.003	0.329
SpO2 in ED	0.025	0.060	0.172	1	0.678	1.025
BE	−0.067	0.036	3.483	1	0.062	0.935
Lactate values	0.155	0.091	2.931	1	0.087	1.168
Demcompressive mini-thoracotomy	0.714	2.619	0.074	1	0.785	2.041
n° rib fractures	−0.082	0.064	1.681	1	0.195	0.921
1st and/or 2nd rib fractures	−1.049	0.424	6.127	1	0.013	0.350
Bilateral rib fractures	0.468	0.551	0.720	1	0.396	1.596
Heart/large vessels injuries	2.570	1.525	2.839	1	0.092	13.071
Pneumothorax	−0.147	0.359	0.167	1	0.683	0.864
Hemothorax	1.201	0.729	2.714	1	0.099	3.325
Hemopneumothorax	0.654	0.553	1.400	1	0.237	1.923
Pulmonary contusion	0.038	0.363	0.011	1	0.916	1.039
Scapula/clavicle/sternum fractures	0.772	0.458	2.839	1	0.092	2.163
Pleural effusion	0.230	0.482	0.227	1	0.634	1.258
Isolated chest trauma	0.020	0.725	0.001	1	0.978	1.020
Abdomen trauma	0.123	0.446	0.076	1	0.782	1.131
Head trauma	0.097	0.467	0.043	1	0.836	1.102
Extremities trauma	0.277	0.433	0.410	1	0.522	1.319
ISS ***	0.088	0.018	22.730	1	<0.001	1.092

*** Significant value.

**Table 4 life-11-01154-t004:** Invasive mechanical ventilation multivariate analysis: anatomical variables.

Invasive Mechanical Ventilation—Anatomical Injuries						
	B	S.E.	Wald	gl	Sign.	Odd Ratio
n° rib fractures	−0.018	0.030	0.360	1	0.548	0.982
1st and/or 2nd rib fractures	0.133	0.182	0.529	1	0.467	1.142
Bilateral rib fractures ***	0.705	0.227	9.615	1	0.002	2.024
Heart/large vessels injuries	0.815	0.616	1.750	1	0.186	2.260
Pneumothorax	0.226	0.176	1.652	1	0.199	1.254
Hemothorax ***	1.016	0.385	6.978	1	0.008	2.762
Hemopneumothorax ***	1.049	0.289	13.123	1	<0.001	2.853
Pulmonary contusion ***	0.674	0.171	15.604	1	<0.001	1.961
Scapula/collarbone/sternum fractures	0.123	0.230	0.288	1	0.591	1.131
Pleural effusion	0.343	0.215	2.542	1	0.111	1.409
Isolated chest trauma	−0.367	0.355	1.068	1	0.301	0.693
Abdomen trauma ***	0.67	0.198	11.465	1	0.001	1.954
Head trauma ***	0.935	0.237	15.508	1	<0.001	2.547
Extremities trauma ***	0.507	0.186	7.442	1	0.006	1.660

*** Significant value.

**Table 5 life-11-01154-t005:** Invasive mechanical ventilation multivariate analysis: physiological variables.

Invasive Mechanical Ventilation—Physiological Parameters						
	B	S.E.	Wald	gl	Sign	Odd Ratio
SpO2 pre-hosp	−0.059	0.038	2.365	1	0.124	0.943
GCS pre-hosp	0.107	0.269	0.160	1	0.689	1.113
SpO2 in ED	−0.018	0.041	0.192	1	0.661	0.982
GCS in ED ***	−1.005	0.254	15.698	1	<0.001	0.366
Systolic blood pressure in ED ***	−0.028	0.006	18.398	1	<0.001	0.973
BE	−0.058	0.033	3.039	1	0.081	0.944
Lactate values ***	0.200	0.080	6.185	1	0.013	1.221

*** Significant value.

**Table 6 life-11-01154-t006:** Non-invasive mechanical ventilation univariate analysis: demographic and trauma-related data.

Non-Invasive Mechanical Ventilation												
		Non NIMV					NIMV					
		n	%	IQR1	Median	IQR3	n	%	IQR1	Median	IQR3	*p*-Value
Gender	Female	128	22.50%				31	17.14%				0.144
	Male	441	77.50%				150	82.87%				
Age				37	48	60			38	50	62	0.254
Age > 65		107	18.80%				42	23.20%				0.201
Age > 75		51	8.98%				20	11.05%				0.387
ASA score	1	364	64.08%				101	55.80%				0.148
	2	173	30.46%				64	35.36%				
	3	30	5.28%				15	8.29%				
	4	1	0.18%				1	0.55%				
RTS pre-hosp ***				12	12	12			11	12	12	0.005
SpO2 pre-hosp ***				96	98	99			94	97	99	0.009
GCS pre-hosp ***				14	15	15			13	15	15	0.078
RTS in ED ***				11.5	12	12			4	12	12	<0.001
SpO2 in ED				97	99	100			96	99	100	0.443
GCS in ED ***				13	15	15			3	15	15	<0.001
Systolic blood pressure in ED ***				115	130	150			100	120	140	<0.001
Mechanisms of injury trauma	other	8	1.45%				2	1.15%				0.869
	stab wound	1	0.18%				0	0.00%				
	gunshot wound	1	0.18%				0	0.00%				
	car accident	113	20.55%				37	21.26%				
	bike accident	38	6.91%				13	7.47%				
	motorcycle accident	176	32.00%				60	34.48%				
	pedestrian accident	88	16.00%				19	10.92%				
	precipitation	120	21.82%				42	24.14%				
	crusching	5	0.91%				1	0.57%				
BE ***				−4.4	−1.9	0.7			−5.3	−3.0	−0.6	0.016
Lactate values ***				1.56	2.225	3.21			1.885	2.63	3.6	0.002
E-FAST ***		110	19.71%				56	31.82%				0.001
Chest X-ray		487	85.59%				148	81.77%				0.236
Torso CT ***		486	85.41%				175	96.69%				<0.001
Decompressive mini-thoracotomy ***		9	1.58%				9	4.97%				0.021
Right chest tube thoracostomy ***		61	10.72%				36	19.89%				0.002
Left chest tube thoracostomy ***		64	11.25%				34	18.78%				0.011
Emergency surgery ***		180	31.63%				86	47.51%				<0.001
Previous intubation ***		180	31.63%				91	50.28%				<0.001

*** Significant value.

**Table 7 life-11-01154-t007:** Non-invasive mechanical ventilation univariate analysis: injuries and trauma score data.

Non-Invasive Mechanical Ventilation												
		Non NIMV					NIMV					
		n	%	IQR1	Median	IQR3	n	%	IQR1	Median	IQR3	*p*-Value
n° rib fractures ***				2	4	6			3	6	8	<0.001
Rib fractures ≤2 ***		209	36.73%				32	17.68%				<0.001
Rib fractures 3–5		216	37.96%				54	29.83%				0.051
Rib fractures ≥6 ***		144	25.53%				95	52.49%				<0.001
1st and/or 2nd rib fractures ***		92	16.17%				53	29.28%				<0.001
Bilateral rib fractures ***		191	33.63%				92	50.83%				<0.001
Heart/large vessels injuries		10	1.76%				7	3.87%				0.146
Pneumothorax ***		199	34.97%				89	49.17%				0.001
Hemothorax ***		19	3.34%				15	8.29%				0.012
Hemopneumothorax		49	8.61%				23	12.71%				0.112
Pulmonary contusion ***		238	41.83%				108	59.67%				<0.001
Scapula/clavicle/sternum fractures ****		78	13.71%				41	22.65%				0.007
Pleural effusion ***		83	14.61%				44	24.31%				0.004
Isoleted chest trauma ***		137	24.08%				22	12.15%				0.001
Abdomen trauma ***		240	42.18%				98	54.14%				0.006
Head trauma		338	59.40%				119	65.75%				0.137
Extremity trauma ***		374	65.73%				142	78.45%				0.001
Chest AIS ***				3	3	3			3	3	4	<0.001
Abdomen AIS				2	2	3			2	2	4	0.379
Head AIS ***				2	2	4			2	3	4	0.003
Extremities AIS ***				2	2	3			2	2	4	0.025
ISS ***				13	19	29			22	29	38	<0.001
Death probability ***				1.1	3.2	10.65			3.00	8.45	20.65	<0.001

*** Significant value.

**Table 8 life-11-01154-t008:** Non-invasive mechanical ventilation multivariate analysis.

Non-Invasive Mechanical Ventilation						
	B	S.E.	Wald	gl	Sign.	Odd Ratio
SpO2 pre-hosp	−0.025	0.022	1.307	1	0.253	0.975
RTS pre-hosp ***	0.202	0.095	4.492	1	0.034	1.224
RTS in ED	−0.039	0.048	0.680	1	0.409	0.961
BE	0.012	0.028	0.176	1	0.675	1.012
Lactate values	−0.005	0.058	0.009	1	0.925	0.995
Decompressive mini-thoracotomy	−0.011	0.645	0.000	1	0.986	0.989
Right chest tube thoracostomy	−0.131	0.324	0.164	1	0.685	0.877
Left chest tube thoracostomy	−0.137	0.329	0.174	1	0.676	0.872
n° rib fractures ***	0.131	0.038	11.851	1	<0.001	1.140
1st and/or 2nd rib fractures	0.242	0.255	0.902	1	0.342	1.274
Bilateral rib fractures	−0.339	0.322	1.098	1	0.295	0.713
Pneumothorax	0.307	0.249	1.514	1	0.219	1.359
Hemothorax ***	1.087	0.529	4.221	1	0.040	2.964
Pulmonary contusion	0.112	0.243	0.211	1	0.646	1.118
Scapula/clavicle/sternum fractures	0.246	0.306	0.645	1	0.422	1.279
Pleural effusion	0.343	0.283	1.472	1	0.225	1.409
Isolated chest trauma	−0.482	0.385	1.566	1	0.211	0.608
Abdomen trauma	−0.278	0.272	1.046	1	0.306	0.757
Extremities trauma	0.151	0.287	0.276	1	0.599	1.163
ISS ***	0.029	0.012	5.566	1	0.018	1.029
Previous intubation	0.035	0.358	0.010	1	0.922	1.036

*** Significant value.

**Table 9 life-11-01154-t009:** Outcome univariate analysis: demographic and trauma-related data.

Outcome												
		Survived					Deceased					
		n	%	IQR1	Median	IQR3	n	%	IQR1	Median	IQR3	*p*-Value
Gender	Female	160	21.19%				19	24.68%				0.469
	Male	595	78.81%				58	75.32%				
Age ***				38	49	60			47	62	78	<0.001
Age > 65 ***		145	19.21%				36	46.75%				<0.001
Age > 75 ***		70	9.28%				24	31.17%				<0.001
ASA score ***	1	472	62.60%				36	46.75%				0.003
	2	237	31.43%				29	37.66%				
	3	43	5.70%				12	15.58%				
	4	2	0.27%				0	0.00%				
RTS pre-hosp ***				11	12	12			3	8	11	<0.001
SpO2 pre-hosp ***				95	98	99			85	95	97	<0.001
GCS pre-hosp ***				14	15	15			3	6	12	<0.001
RTS in ED ***				11	12	12			3	4	4	<0.001
SpO2 in ED ***				97	99	100			90	98	100	0.014
GCS in ED ***				10	15	15			3	3	3	<0.001
Systolic blood pressure in ED ***				110	130	146			75	90	120	<0.001
Mechanisms of injury trauma	other	10	1.37%				1	1.33%				0.110
	stab wound	1	0.14%				0	0.00%				
	gunshot wound	1	0.14%				0	0.00%				
	car accident	151	20.74%				10	13.33%				
	bike accident	51	7.01%				6	8.00%				
	motorcycle accident	240	32.97%				16	21.33%				
	pedestrian accident	107	14.70%				14	18.67%				
	precipitation	161	22.12%				28	37.33%				
	crusching	6	0.82%				0	0.00%				
BE ***				−4.5	−2.1	0.5			−12	−6.6	−3.1	<0.001
Lactate values ***				1.605	2.295	3.235			2.94	5.31	10.1	<0.001
E-FAST		167	22.63%				21	29.17%				0.241
Chest X- ray ***		641	84.90%				50	64.94%				<0.001
Torso CT		663	87.81%				69	89.61%				0.854
Decompressive mini-thoracotomy ***		17	2.25%				11	14.29%				<0.001
Right chest tube thoracostomy ***		98	12.98%				17	22.08%				0.036
Left chest tube thoracostomy ***		96	12.72%				22	28.57%				<0.001
Emergency surgery ***		266	35.23%				46	59.74%				<0.001
Previous intubation ***		265	35.10%				71	92.21%				<0.001

*** Significant value.

**Table 10 life-11-01154-t010:** Outcome univariate analysis: injuries and trauma score data.

Outcome												
		Survived					Deceased					
		n	%	IQR1	Median	IQR3	n	%	IQR1	Median	IQR3	*p*-Value
n° rib fractures ***				2	4	6			3	6	9	<0.001
Rib fractures ≤2 ***		244	32.32%				12	15.58%				0.002
Rib fractures 3–5		271	35.89%				25	32.47%				0.618
Rib fractures ≥6 ***		240	32.00%				40	51.95%				0.001
1st and/or 2nd rib fractures ***		284	37.67%				38	49.35%				0.050
Bilateral rib fractures ***		147	19.47%				33	42.86%				<0.001
Heart/large vessels injuries		16	2.12%				3	3.90%				0.408
Pneumothorax		292	38.68%				34	44.16%				0.391
Hemothorax ***		34	4.50%				8	10.39%				0.048
Hemopneumothorax		72	9.54%				12	15.58%				0.11
Pulmonary contusion ***		346	45.83%				48	62.34%				0.006
Scapula/collarbone/sternum fractures		120	15.89%				14	18.18%				0.625
Pleural effusion		125	16.58%				17	22.08%				0.264
Isolated chest trauma ***		164	21.72%				2	2.60%				<0.001
Abdomen trauma ***		338	44.77%				48	62.34%				0.004
Head trauma ***		454	60.13%				74	96.10%				<0.001
Extremity trauma		516	68.34%				57	74.03%				0.366
Chest AIS ***				3	3	4			3	3	4	<0.001
Abdomen AIS ***				2	2	3			2	2	4	<0.001
Head AIS ***				2	2	4			3	5	5	<0.001
Extremities AIS ***				2	2	3			2	4	5	<0.001
ISS ***				14	22	32			38	43	54	<0.001
Death probability ***				1.3	4.1	12.4			39.1	83.2	96.8	<0.001

*** Significant value.

**Table 11 life-11-01154-t011:** Outcome multivariate analysis.

Outcome						
	B	S.E.	Wald	gl	Sign.	Odd Ratio
Age > 65 ***	1.638	0.373	19.331	1	<0.001	5.146
ASA			2.180	3	0.536	
ASA 1	−0.365	0.363	1.011	1	0.315	0.694
ASA 2	0.192	0.483	0.159	1	0.690	1.211
ASA 3	−20.263	25089	0.000	1	0.999	0.000
n° rib fractures	−0.012	0.041	0.092	1	0.761	0.988
Bilateral rib fracures ***	1.082	0.321	11.348	1	0.001	2.952
Hemothorax ***	1.142	0.484	5.578	1	0.018	3.133
Pulmonary contusion ***	0.620	0.286	4.685	1	0.030	1.858
Pleural effusion	0.033	0.331	0.010	1	0.920	1.034
Isolated chest trauma	0.738	1.053	0.491	1	0.483	2.091
Abdomen trauma	0.499	0.285	3.078	1	0.079	1.648
Head trauma ***	3.163	0.788	16.110	1	<0.001	23.638

*** Significant value.

## Data Availability

The data presented in this study are available on request from the corresponding author. The data are not publicly available due to privacy.

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
