# Peer review of "Predictive Factors of Ventilatory Support in Chest Trauma"

_life, 2021, doi:10.3390/life11111154_

Round 1
Reviewer 1 Report
Congratulations on a well-prepared publication. The material and method are prepared in a stationary manner, similarly the results and their presentation in the tables are excellent.Major Concern:
1. however, the discussion seems to be the weakest part of the manuscript. In discussions, authors often repeat their results from the results. as a reader, I would expect more discussion and references to key publications.
2. Literature collection could be better:
Rico FR, Cheng JD, Gestring ML, Piotrowski ES. Mechanical ventilation strategies in massive chest trauma. Crit Care Clin. 2007 Apr;23(2):299-315, xi. doi: 10.1016/j.ccc.2006.12.007. PMID: 17368173.
3. We no nothing about types of used ventilation: one lung or double lung strategy; support in trachea-bronchial injury?
Minor:
linę 115 - should be twenty-eight
Limitation can be seperated chapter.
Author Response
Thank you for your comments and suggestions. I corrected line 115, separated the limitations of the study and I tried to improve the discussion as you suggested.
I read the article that you suggest and I would like to answer your question. Thoracic trauma is an evolving condition. Sometimes, the need of respiratory support is initially underestimated, mostly in absence of associated conditions requiring orotracheal intubation. In this case pulmonary function progressively deteriorates. On the other hand, a timely use of non-invasive ventilation in patients with no criteria for mechanical ventilation, but requiring positive pressure to improve oxigenation in case of pulmonary contusions may avoid clinical impairment, improving the clinical outcome.
The aim of our study was to assess which ventilatory support was indicated based of the conditions of the patient. We did not consider independent lung ventilation and extracorporeal lung ventilation since they are a very invasive and complex treatments that occurs in particular severe conditions and none of our patients needed it.
Reviewer 2 Report
Your purpose is to define the predictive indicators of the need for ventilator support in patients with thoracic trauma.
General: not academic scientific writing.
:Please add a confidence interval (CI) with odds ratio.
:discussion, at first paragraph, describe the summary of your results.
:please write your limitation clearly through your research. For example, selection bias, information bias, confounding factor,
: please write your strength points and clinical implication clearly.
Minor points
Line 9; abstract; remove the citation [1].
Line 16: “multivariable analysis” is correct than “multivariate analysis”?
Line 65:what is the indicators for ventilator support in the clinical setting?
Author Response
Thank you for your comments and suggestion. I added CI to Odd Ratio, a summary of results in the discussion and pointed out clearly the limitation and the strengths of this study, finally, I removed the citation [1] in the abstract.
Line 16. No, it isn’t. Multivariate analysis refers to the statistic analysis we performed to investigate which variables are indicators of the need of orotracheal intubation
Line 65. This line refers to the “material and methods” paragraph. The indicators of ventilation in the clinical setting are the results we aimed to investigate. To date, not considering the results of the present study, in our daily clinical practice the patients requiring mechanical ventilation are those with severe head trauma, shock conditions, respiratory insufficiency apart of the number of rib fractures and those having surgery. We hope that our study could improve a better selection of patients requiring ventilatory support.
Reviewer 3 Report
"Twenty-height patients" - need correction.
Author Response
Thank you for your suggestions. I corrected "twenty-eight".
Round 2
Reviewer 2 Report
:method: in your clinical setting, describe the indication of ventilation clearly.
:variable: For example, hemothorax and hemopneumothorax is near disease concept.
In multivariable analysis, did the condition of multicollinearity meet?
:discussion, at first paragraph, instead of bullet points, describe the summary of your results as logical scientific writing..